# Investigating the Interaction between Negative Strand RNA Viruses and Their Hosts for Enhanced Vaccine Development and Production

**DOI:** 10.3390/vaccines9010059

**Published:** 2021-01-17

**Authors:** Kostlend Mara, Meiling Dai, Aaron M. Brice, Marina R. Alexander, Leon Tribolet, Daniel S. Layton, Andrew G. D. Bean

**Affiliations:** CSIRO Health & Biosecurity, Australian Centre for Disease Preparedness, Geelong, Vic 3220, Australia; meiling.dai@csiro.au (M.D.); aaron.brice@csiro.au (A.M.B.); marina.alexander@csiro.au (M.R.A.); leon.tribolet@csiro.au (L.T.); daniel.layton@csiro.au (D.S.L.); andrew.bean@csiro.au (A.G.D.B.)

**Keywords:** NSV vaccines, RNA viruses, host immune response, host–pathogen interaction, vaccine design, vaccine manufacturing

## Abstract

The current pandemic has highlighted the ever-increasing risk of human to human spread of zoonotic pathogens. A number of medically-relevant zoonotic pathogens are negative-strand RNA viruses (NSVs). NSVs are derived from different virus families. Examples like Ebola are known for causing severe symptoms and high mortality rates. Some, like influenza, are known for their ease of person-to-person transmission and lack of pre-existing immunity, enabling rapid spread across many countries around the globe. Containment of outbreaks of NSVs can be difficult owing to their unpredictability and the absence of effective control measures, such as vaccines and antiviral therapeutics. In addition, there remains a lack of essential knowledge of the host–pathogen response that are induced by NSVs, particularly of the immune responses that provide protection. Vaccines are the most effective method for preventing infectious diseases. In fact, in the event of a pandemic, appropriate vaccine design and speed of vaccine supply is the most critical factor in protecting the population, as vaccination is the only sustainable defense. Vaccines need to be safe, efficient, and cost-effective, which is influenced by our understanding of the host–pathogen interface. Additionally, some of the major challenges of vaccines are the establishment of a long-lasting immunity offering cross protection to emerging strains. Although many NSVs are controlled through immunisations, for some, vaccine design has failed or efficacy has proven unreliable. The key behind designing a successful vaccine is understanding the host–pathogen interaction and the host immune response towards NSVs. In this paper, we review the recent research in vaccine design against NSVs and explore the immune responses induced by these viruses. The generation of a robust and integrated approach to development capability and vaccine manufacture can collaboratively support the management of outbreaking NSV disease health risks.

## 1. Introduction

The human population is under the constant threat from infectious diseases, as seen in the consistent sporadic Ebola virus epidemics and the recent coronavirus and influenza virus pandemics. There is no doubt that the rapid global spread of viruses can have significant and far-reaching impact on health systems and the world’s economy. Research in emerging infectious diseases is advancing rapidly, with new breakthroughs in the understanding of host–pathogen interactions and the development of innovative and exciting vaccination strategies [1]. Since pandemics present such all-encompassing catastrophes, we must invest in developing basic research fundamentals and a better understanding of host–pathogen interactions for improved vaccine production to protect us from currently circulating infectious diseases and enable rapid response to emerging threats. As such, a significant challenge to achieve this goal is to refine the tools and processes necessary to efficiently develop and produce efficacious vaccines.

RNA viruses cause up to 44% of all emerging infectious diseases [2]. Negative strand RNA viruses (NSVs; order *Mononegavirales*) in particular are widely disseminated and of significant concern to human and animal health. These include the virus families *Paramyxoviridae* (measles (MV), mumps (MuV), respiratory syncytial virus (RSV), and human parainfluenza (HPIV) viruses), *Orthomyxoviridae* (influenza A virus (IAV)), *Rhabdoviridae* (rabies virus (RABV) and vesicular stomatitis virus (VSV)), *Filoviridae* (Ebola (EBOV), and Marburg (MARV) viruses). These viruses are responsible for a high burden of morbidity and mortality, especially in the developing world. Unlike positive strand RNA viruses, which are immediately translated by the host cells, thus directly facilitating replication and spread within the host, NSVs need to first have their RNA transcribed into a positive strand before initiating replication [3]. Due to their smaller genome size compared to DNA viruses, RNA viruses rely more heavily upon host cellular proteins [4]. However, this is hampered by their shorter generation time and the lack of polymerase proofreading function, leading to higher rates of mutation, up to five orders of magnitude compared to some DNA viruses [5]. This additional allows RNA viruses to more readily infect new host species [6]. Finally, the lack of effective animal models and the requirement of high containment facilities to perform NSV research contribute to the challenges of studying NSVs.

Some of the diseases caused by NSVs have been successfully controlled through immunization, as in the case of MV and MuV. Nevertheless, others still require the development of appropriate strategies for long term vaccination successes. These include both traditional and innovative strategies, involving live attenuated viruses, inactivated, subunit, protein, vectored, and nucleic acids vaccines [7,8,9,10]. Additionally, research has not only focused on the vaccine itself, but also on the procedures associated with their manufacture to improve the speed, yield, and cost of production. For example, methods using vaccine production substrates that incorporate approaches from the use of embryonated chicken eggs, baculovirus expression vectors in insect cells, and synthetic chemistry and use of engineered human or animal cells [6,7,8,9,10].

Nonetheless, the success of a vaccine stands not only on its manufacturing efficiency, but also on its ability to induce long lasting protective immunity. Vaccines present a range of different antigens to the cells located both on the surface and within virus particles, and it is critical that they induce appropriate responses to generate immune memory. As such, it is vital to understand host–pathogen interactions to ensure that vaccines are developed to stimulate the most appropriate and protective response for a long-lasting solution to NSVs. Moreover, given that NSVs have a multitude of ways in which they can co-opt or manipulate the host’s immune system to benefit viral growth and spread, it is additionally important to understand this interface to optimize treatment options and improve vaccine design and manufacture.

The aim of this review is to summarize current achievements in the design of NSV vaccine approaches and highlight new tactics for vaccine strategy improvement. In this review, we will focus on some NSV, which have an impact on human health and whose immunisation strategies have for some been successful, but nevertheless for others remain a challenge. We discuss the types of host immune response induced by NSVs, followed by how NSVs escape this host immune response. Additionally, we present some examples of success and failures of NSV vaccines highlighting the factors behind these achievements. Furthermore, an overview of current NSV vaccine design platforms illustrating some examples of vaccines already on the market or undergoing clinical trials is discussed. Finally, we discuss the challenges to the manufacturing of NSV vaccines.

## 2. Host Factors that Affect NSV Vaccine-Induced Immunity

During the NSV infection, viral conserved components called pathogen associated molecular patterns (PAMPs) are recognized by host pathogen recognition receptors (PRRs), such as the retinoic acid-inducible gene-I (RIG-I) and toll-like receptors (TLRs) [11]. The activation of these innate sensing pathways eventually leads to the production of IFNs and cytokines/chemokines critical for efficient activation of adaptive immune responses (B- and T-cell responses) that help control and clear infection and produce immunological memory to rapidly respond to future infection. The detection of viral nucleic acid leads to the activation of latent transcriptions (IRF3, IRF7, and NFκB) and expression of type-I IFNs (IFNα and IFNβ) and proinflammatory cytokines (such as IL6 and TNFα) [12]. Type-I IFNs then act in an autocrine or paracrine fashion to stimulate the expression antiviral ISGs via STAT1/2 [12] (Figure 1).

Therefore, the quality and magnitude of adaptive immunity is dependent on the innate immune response [13]. Supporting this, Nakaya et al. found that antibody titers at one-month post vaccination were positively correlated with early expression of type I IFNs associated genes [14]. Early induction of IFN was also reported to be important for the development of antibody responses in LAIV (live attenuated influenza vaccines) and TIV (trivalent influenza vaccines) vaccinated children [15]. TLRs have been shown to play an important role in both cell-mediated and antibody mediated protection in measles vaccine response. Secretion of type-I IFNs promotes the recruitment and activation of specialised immune cells that aid in the clearance of viral pathogens. Of this cell-mediated immune response, dendritic cells are central to bridging innate and adaptive immunity through their functions as antigen-presenting cells, leading to the activation T-helper cells that, in turn, activate humoral immunity via the production of a pathogen-specific antibody by B-cells [16]. Robust activation of T- and B-cells also leads to the generation of immune memory, a key requirement for vaccines to induce long-lasting immunity [17] (Figure 1). The importance of specific innate cell subsets such as natural killer (NK) and T follicular helper cells has been observed as correlates of protection in recipients of Ebola vaccine. Additionally, polymorphisms in dendritic cell-specific intercellular adhesion molecule-3 grabbing nonintegrin (DC-SIGN), a measles-specific receptor, may modulate cytokine responses to the measles component of the MMR vaccine [18]. Although the adaptive immune response has traditionally been the primary focus of vaccine development, these studies have highlighted the importance of innate immune responses after vaccination and infection. Thus, strategies can be developed to trigger specific innate pathways that can lead to stronger adaptive immune responses which are protective against infection.

T-cells and B-cells are critical components in adaptive immunity against IAV infection. CD4^+^ T-cells target IAV-infected epithelial cells through binding to MHC class II molecules and contribute to B cell activation promoting antibody production [19]. CD8^+^ T-cells differentiate into cytotoxic T lymphocytes (CTLs) and defend against IAV infection via producing cytokines and effector molecules in addition to direct cytotoxic effects on infected cells mediated by MHC class I [20]. Since cytotoxic T-cells target infected cells, and not the virus directly, T-cell mediated immunity typically provides protection without complete virus neutralization, allowing some degree of virus replication [21]. T-cell recognition of conserved viral antigens presented by antigen presenting cells (APC) can contribute to a more qualitative antibody response. However, current inactivated vaccines do not elicit a strong T-cell response, which may in part be overcome by the use of optimal adjuvants [22]. Polymorphisms in the signaling lymphocyte activation molecule (SLAM) and CD46 have been shown to drastically reduce MV-specific antibodies in individuals’ post-vaccination, which illustrates the importance of genetic variation in our immune cell surface markers on vaccine responses [23]. A correlation in the number of plasmacytoid dendritic cells and MV-specific antibodies has also been observed post-vaccination in infants, indicating another important link between the adaptive and innate immunity [24]. Induction of antibodies targeting conserved IAV antigens is desirable for vaccine responses with the potential to provide broad protection. Virus-neutralizing antibodies usually target viral surface antigens, for example the IAV haemagglutinin (HA) head domain. However, multiple immune mechanisms that do not result in virus neutralization can also contribute to heterosubtypic immunity against different subtypes of influenza viruses [25]. Ways by which the humoral branch of the vaccine response can also contribute to protection other than virus neutralization is through Fc receptor engagement via antibody-dependent cellular phagocytosis (ADCP) [26,27] or antibody-dependent cellular cytotoxicity (ADCC) [28,29,30], as well as antibody-dependent complement mediated-lysis (ADCL) [31]. Additionally, external factors that can modulate the effectiveness of the adaptive immune system, such as post-transplant immunosuppressant medication, can lead to a reduction in B-cells and protective antibody responses to MV over time [32].

## 3. Immune Evasion by NSVs

As mentioned in Section 2, viruses must overcome robust immune defences in order to establish productive infection. The NSVs discussed in this manuscript expertly evade these responses through a multitude of mechanisms summarised in Figure 1 and reviewed elsewhere [33,34,35,36,52]. Understanding how these viruses evade these immune defenses, especially the host–pathogen interfaces involved, are integral to the development of efficacious vaccine strains both by reducing pathogenicity and ensuring formation of immunological memory. In this section, we highlight a few points of convergence where these NSVs target our immune responses and the implications for vaccine development.

The IFN response is the primary defense against viral infection. Two stages of this response that are commonly targeted by NSVs are proverbial ‘bottlenecks’, viral RNA sensing and the ISGF3 transcription complex (consisting of STAT1, STAT2, and IRF9). Viral RNA is detected by two families of pattern recognition receptors (PRRs), RIG-I-like receptors (RLRs), and toll-like receptors (TLRS). The RLRs RIG-I and mda-5 detect uncapped RNA and double-stranded RNA (dsRNA), respectively, in the cytoplasmic compartment [53]. Toll-like receptors (TLRs) instead sense viral RNA within the endosomal compartment, these include TLR3, which detects dsRNA, and TLR7 and 8, which detected single-stranded RNA (ssRNA) [54]. RLRs are the principal PRRs that detect the NSVs discussed in this review [53] and, as such, each has a mechanism evading detection. In some cases, this is believed to be a ‘passive’ mechanism whereby the viral RNA is hidden from RLRs, such as by encapsidation by the RABV N protein and sequestration by EBOV VP35 [55,56]. RSV NS2 and MV V protein actively associate with RIG-I or MDA5, respectively, to inhibit its activity and RSV N sequesters MDA5 in viral inclusion bodies [51,57,58]. MV V also targets the phosphatases PPIα/γ and IAV NS1 targets the ubiquitin ligase TRIM25, both preventing their functions in the activation of RIG-I [59,60].

ISGF3 is responsible for the expression of hundreds of IFN-stimulated genes (ISGs), many of which have antiviral activity [61]. Thus, it is not unexpected that components of this complex, in particular STAT1, are common targets for viral IFN antagonists. The *P* gene products of both RABV (P1–P5) and MV (P, V, C) variously inhibit STAT1 function. RABV P1 binds to ‘active’, phosphorylated STAT1 (pY-STAT1) and excludes it from the nucleus, while the smaller P3 tethers pY-STAT1 to cellular microtubules to inhibit its nuclear import and additionally blocks its DNA-binding intranuclearly [62,63,64]. MV P associates with unphosphorylated STAT1 (U-STAT) to inhibit its phosphorylation and subsequent nuclear accumulation, and the V protein prevents phosphorylation of Tyk2, in addition to associating with Jak1, STAT1, and STAT2, to contribute to inhibition of ISGF3 activation and nuclear import [65,66,67,68]. The C protein instead targets pY-STAT1, inhibiting correct formation of STAT1 homodimers in response to the type-III IFN, IFNγ [69]. In addition, the MV N protein also contributes to this STAT1 blockade by inhibiting the nuclear accumulation of pY-STAT1 [70]. ISGF3 targeting by RSV, EBOV, and IAV are less multifaceted, RSV NS1 and NS2 target STAT2 for degradation, and IAV upregulates the expression of SOCS1 and SOCS3 that negatively regulated STAT signaling by dephosphorylating/inactivating JAK1 and TYK2 [71,72,73,74]. Finally, EBOV VP24 inhibits pY-STAT1 nuclear accumulation by competitively binding with the nuclear transport proteins importin-α1, α5, and α6 [75,76]; VP24 has also been found to bind to U-STAT1, but the contributions of this interaction to immune evasion are not yet known [77]. Importantly, inhibition of STAT1 contributes to the pathogenicity of all five viruses, evaluated through infection of STAT1 knockout mice or using mutant virus unable to target STAT1 [72,78,79,80,81]. Thus, disruption of the virus-STAT1 interface is central to the production of attenuated vaccines.

The NSVs discussed in this manuscript commonly target T-cells to prevent their antiviral immune functions and their ability to activate humoral immunity. During RABV infection, only transient infiltration of T-cells into the CNS is observed, this is believed to be due to the depletion of T-cell populations via upregulation of the immunosuppressive molecules FasL, B7-H1, and HLA-G on infected neurons [82,83,84]. MV inhibits the proliferation of uninfected T-cells at the cell surface involving the surface exposed GP complex, consisting of the HA and F proteins, though the specific mechanism is yet to be elucidated [85,86,87]. EBOV has similarly been shown to induce apoptosis of T-cells through interactions of its surface GP with TLR4 [88]. EBOV VP40 is also released from infected cells in exosomes, which have also been proposed to induce apoptosis in uninfected T-cells. While not targeting T-cells specifically, RSV inhibits the activation of T-cells by DCs via N protein, which localises to the surface of DCs and impairs formation of the immunological synapse through reduction of MHC clustering [89]. IAV also does not target T-cell directly, but its high mutation rate in immunological epitopes of its N protein have been shown to prevent recognition of infected cells by CD8+ T-cells [45]; this could possibly extend to evasion of CD4+ T-cells as well. In addition, IAV, RSV, and EBOV all suppress DC maturation, inhibiting their capacity to activate T-cells, while MV does not impact maturation, but the capacity for infected DCs to activate T-cells is greatly reduced [90,91,92,93,94].

## 4. Factors behind the Success/Failure of an NSV Vaccine

Many of the diseases caused by NSV are prevented by immunizations. However, for some of these viruses, the efforts made to design a vaccine have failed or the available vaccines have induced a suboptimal protection. An important key factor in designing a successful vaccine is to use previous experience on existing ones. In this section we will discuss some of the factors that have influenced the success or failure of an NSV vaccine with focus on five major NSV causing diseases.

### 4.1. Measles

MV belongs to the *Paramyxovirus* family and causes a highly contagious disease in children, characterised by month-long immune depression in 30% of cases and a fatal long-term complication of subacute sclerosing panencephalitis (SSPE) in up to 0.01% of cases [48,95]. Excluding reduced uptake through anti-vaccine campaigns, the measles vaccine has been one of the most successful, preventing an estimated 52 million cases in 20 years in the US alone [96] with an excellent safety profile. MV is an antigenically invariant pathogen where acute infection elicits protective immunity in convalescent persons. This allowed the licensing of a live virus attenuated vaccine via passage of a human isolate through tissue culture [97]. Vaccination provides life-long immunity after two inoculations during childhood. Strong cellular and antibody responses are developed by activation of T-cells via TLRs through viral replication and antibody producing B-cells. Supporting this, gene polymorphisms in TLRs 2, 3, 4, 6, 7, and 8 influenced antibody levels, neutralization, and cell-mediated responses in 764 vaccine recipients [98], emphasising the importance of these innate immune receptors in effective vaccine responses. Attenuation of the vaccine strain by tissue culture passage may be due to earlier type I IFN induction triggered by the presence of defective interfering RNA residing inside virus particles that signal through TLR3/4 within hours post infection [99]. It is interesting to speculate that in the absence of whole organism adaptive immune pressure the deleterious effect of early IFN responses on virus survival or persistence is less, allowing these characteristics to re-emerge. Clearly the flow-on signaling effects of type I IFNs on eventual viral clearance by T- and B-cell responses is a significant host control measure for NSVs [100].

### 4.2. Influenza

Pandemic influenza outbreaks occur on a periodic basis with the most recent being the 2009 H1N1 pandemic. Annual influenza epidemics cause 3–5 million severe cases of respiratory illnesses and 290,000–650,000 deaths worldwide [101], posing a tremendous impact on global health. Vaccination is the primary strategy for the prevention and control of influenza. Annual influenza vaccination is currently recommended for groups at high risk of complications from influenza infection such as elderly people, young children, people with chronic diseases, and pregnant women, as well as health workers [102].

Current licensed influenza vaccines include either inactivated or live attenuated influenza viruses. Inactivated vaccines are either split virus, subunit vaccines, or recombinant HA based vaccines (discussed below) and contain the viral major surface glycoprotein hemagglutinin (HA) from a particular stain. Protection is limited to the induction of vaccine strain specific antibody responses [103]. Live attenuated influenza vaccines provide a multifaceted immune response with local and systemic antibody and T cell responses, but are only available for non-pregnant individuals between 2 and 49 years of age [104].

Current seasonal influenza vaccines have suboptimal effectiveness across all age groups with vaccine effectiveness ranging between 19–60% in US from 2009–2019 [105]. Licensed seasonal influenza vaccines are formulated to induce neutralizing antibodies against strain-specific HA on the viral surface. HA of the circulating virus occasionally undergoes antigenic drift/shift and antigenically mismatches the vaccine strains. Therefore, vaccine-induced antibodies may have reduced effectiveness in targeting the circulating virus, impairing vaccine efficacy. Mutations in the HA as a result of egg-adaptation have also been reported and can result in suboptimal vaccine effectiveness [106]. To overcome this potential production issue, several new influenza vaccine cell culture-based production methods have been granted commercial license in recent years, such as Flublok. Trial data have demonstrated that the higher antigen content in Flublok results in improved immunogenicity and an improved efficacy compared with standard inactivated influenza vaccines [107]. Moreover, this method of vaccine manufacture has the potential to accelerate vaccine production compared to traditional methods in the event of an influenza pandemic or vaccine supply shortage.

Another weakness of current influenza vaccines is that they often fail to induce protective immune responses with a single vaccination in the very young and elderly. Young children typically have no pre-existing immunity against influenza virus, and two consecutive influenza vaccinations are required to induce protective antibody levels. In the elderly, although having pre-existing immunity against influenza, both innate and adaptive immune responses are affected by immunosenescence [108]. To overcome this, an unadjuvanted high-dose vaccine has been developed and shown to reduce inefficient induction of influenza-specific antibodies in the elderly population [109,110]. Adjuvants have also been shown to improve vaccine response, with several approved for use in influenza vaccines, including MF59, AS03, and AF03 [110,111]. As immunostimulatory agents, adjuvants not only enhance the immunogenicity of the co-administered antigens, but potentially accelerate vaccine-induced immune response and increase immunogenicity in populations with poor immune responses, such as the elderly and immunosuppressed patients.

A major challenge for vaccine development for IAV viruses is that they continually undergo antigenic variation, and both inactivated and live attenuated influenza vaccines confer only short-lived strain-specific immunity. Therefore, the development of a highly efficacious universal vaccine to elicit long-lasting and broadly protective responses against antigenically variable viruses would be ideal. Attractive approaches are based on conserved protein regions or peptides (both B- and T-cell epitopes in the same formulation), shared by all strains, and are able to induce cross-protective neutralizing immunity against conserved viral antigens [112]. For example, strategies are being developed to allow strong antibody responses to the HA stem region, the less immunogenic region of the HA, including the use of headless HA, HA molecules with hyper-glycosylated heads, or sequential immunization with chimeric HA molecules [113,114].

Incorporation of diverse and more highly-conserved antigens such as neuraminidase (NA) [115], matrix protein 2 ectodomain (M2e) [116], and nucleoprotein (NP) [117] have also been suggested as potential antigens that could benefit a universal IAV vaccine, in addition to improving the humoral immune response to the HA antigen. Additionally, the internal proteins such as M1, NP, and PB1 show higher degrees of conservation among IAV and contain conserved T cell reactive regions, which are often targeted by the antigen-specific CD8^+^ cytotoxic T-cells and CD4^+^ helper T lymphocytes. In this regard, several universal influenza vaccine candidates in clinical trials target T cell epitopes for improving influenza vaccine efficacy and increasing the breadth of protection against IAVs [118,119]. These approaches are aimed at inducing long-lived, ideally lifelong, immunity. Thus, vaccines that induce both humoral and cell-mediated immune responses directed at conserved regions of IAV, in addition to strain-specific antibodies, are likely to afford broad protective immunity to different IAVs, including drift variants as well as novel virus subtypes.

### 4.3. Rabies

RABV is the best-known member of the *Lyssavirus* genus, the causative agents of the neuropathological disease, rabies, which is inevitably fatal following the onset of symptomatic infection. Vaccination for rabies virus (RABV) has been available for over 135 years, since Louis Pasteur used an emulsion of dried spinal cord from a rabies virus-infected rabbit to successfully treat a nine-year old boy bitten by a rabid dog [120]. Despite such an early success, rabies disease still exists today as a neglected health threat that results in over 55,000 deaths annually, though this number is likely unrepresentative due to its high prevalence in rural areas lacking proper health surveillance [121,122].

Current vaccines for the prevention of rabies are derived from either cell culture or embryonated eggs (purified duck embryo), collectively termed CCEEVs [123]. The purpose of these vaccines is to generate strong Th1 and B cell responses, and subsequent immunological memory, to promote the production of virus neutralising antibodies (VNAs) that are essential for RABV clearance [124,125,126]. The current World Health Organisation (WHO) vaccination recommendation for rabies includes two doses, at day 0 and day 7; this is referred to as pre-exposure prophylaxis (PrEP) [123]. Although this regimen is believed to produce lifelong immunological memory, it does not necessarily provide protection to future challenge in the absence of post-exposure prophylaxis (PEP, discussed below), especially if VNA titres fall below protective levels, which can occur as early three months post-vaccination [79]. Thus, it is recommended that individuals at constant risk of exposure, such as medical and animal workers, undergo antibody monitoring, and booster vaccination provided titres fall below protective levels as an additional precaution against ignorant exposure [123].

As the therapeutic window for rabies ends following onset of symptoms, leading to c. 100% fatality, PEP is still mandated following possible exposure in previously vaccinated individuals, in case of inadequate vaccine protection [123]. PEP for unvaccinated individuals involves three to four doses of vaccine over a one-week or three- to four-week period, depending on the regimen used [79]. Additionally, rabies immunoglobulin (RIG) is provided as soon as possible to provide protection until a vaccine response is generated [123]. For previously vaccinated individuals, the PEP is reduced to a single vaccine dose or two within a three-day period depending on regimen used [123]. RIG is not recommended for vaccinated individuals due to a rapid response to vaccine boosters [123,127].

Despite the availability of a safe and efficacious vaccine, it is not ideally suited for areas where rabies is endemic, namely rural areas of Asia and Africa where 95% of global cases occur [122]. The requirement of cold storage, parenteral administration, serological surveillance, and necessity of multidose PEP following every suspected exposure would necessitate the presence of dedicated medical professionals and infrastructure close to rural settlements to maintain adequate protection, a difficulty in areas of low socioeconomic status [122]. PrEP/PEP is also cost-prohibitive, costing c. $45 USD per patient, raising to c. $100 USD if RIG is also required. Considering the average daily wage in rabies-endemic areas is c. $2 USD, seeking PEP is a considerable financial burden, compounded by the additional loss of several day’s wages due to the multidose regimen [122]. A final consideration is that current vaccines are not protective against all lyssaviruses and only cross-reactive with other members of phylogroup I, leaving six (consisting of phylogroup II and unclassified) out of sixteen species without any form of treatment [128].

A more cost-effective approach to rabies prevention is via mass-vaccination of the animal reservoirs that maintain the virus [122,123]. This has led to the development of oral vaccine baits containing live-attenuated virus that have been used successfully to control rabies in areas where it is maintained in terrestrial mammals other than dogs, such as foxes and raccoon dogs, predominately in Europe [129,130,131]. Oral vaccination of dogs, which account for c. 99% of human rabies cases [132], requires additionally safety consideration due to their close association with humans. Thus, parenteral vaccination using inactivated virus is preferred but requires local populace involvement to bring animals to veterinary clinics or dedicated vaccinators to engage wild dogs, making it difficult to achieve the goal of 70% population coverage believed necessary for rabies eradication [123,132]. Significant investment in dog population surveillance is also required to achieve this goal [122].

Due to the various issues, there remains a space for the development of cheaper, safe, and more efficacious rabies vaccines. One avenue of research involves the use of reverse genetics systems to modify virus gene expression to improve the immunogenicity of currently available non-pathogenic RABV strains, such as ERA, SAD, and RC-HL. Examples include the introduction of host genes such as B-cell activating factor (BAFF) to promote B-cell activity, IFN-α to induce antiviral immune signalling, or flagellin to activate dendritic cells [133,134,135]. The addition of multiple attenuated G-protein-encoding genes, a major contributor to RABV pathogenesis and the primary target of VNAs, has also been shown to reduce pathogenicity and improve immunogenicity [136,137,138,139]. Gene knockout mutants have also been developed to improve safety be preventing generation of virus progeny, this includes matrix (M) or phosphoprotein (P) deficient strains [140,141,142]. Alternative methods to improve safety either substitute the virus for a harmless pseudotyped adenovirus or vaccinia virus expressing the RABV G-protein, or remove the virus entirely by means of DNA vaccines expressing G protein [143,144,145].

In summary, the issues with rabies vaccination do not stem from the lack of a safe and effective vaccine in humans, but due to its poor suitability for the areas where rabies is endemic. The c. 100% fatality rate of symptomatic infection has mandated added precautionary treatment measures that further increase its medical and economic burden and, necessarily, hampered the development of highly immunogenic live-attenuated vaccines. Due to the successful eradication of rabies in many countries, focus has shifted to the development of an efficacious oral vaccines for the main zoonotic reservoir, wild dogs. Nevertheless, the maintenance of numerous lyssaviruses in bats, several of which currently lack any form of therapeutic intervention, highlight the need for the development of pan-lyssavirus vaccines and antiviral therapies able to treat symptomatic infection.

### 4.4. Ebola

EBOV belongs to the *Filovirus* family causing a rare but severe disease with a high case fatality rate (25–90%). It is endemic to central Africa and poses a global threat through imported infections or biological terrorism. Fatal infection is characterised by immune suppression and systemic inflammation that inhibits vascular coagulation, leading to multiorgan failure and shock, with less than half of cases progressing to hemorrhage. Death typically occurs between day 6 and 16 post-infection [146], while non-fatal cases usually present with fever and improve by day 6–11. Survivors show early strong innate responses, high specific IgM responses and cytokines produced by T cells. In contrast, IFN-ɣ, IFN-α, IL-12, 1L-10, and tumor necrosis factor (TNF)-α were associated with death from hemorrhagic fever (for review see [147]). A licensed vaccine has only recently been made available despite pre-clinical work beginning in 2005 when a live-attenuated VSV carrying the EBOV glycoprotein (rVSV-ZEBOV) successfully protected nonhuman primates against EBOV challenge, weeks after immunisation [148]. In a prime boost vaccination strategy, where DNA plasmids are used to prime, subsequently followed by AdV vaccination to boost the response, DNA prime-AdV boost strategies had taken several months [149]. Until the 2013–2016 outbreak in West Africa, it was not considered worth the $1 billion USD to bring to market. Amidst the peak of the outbreak, the WHO and BARDA contracted Merck to scale production of the VSV vectored vaccine. Randomised and blinded phase three trials were conducted in less than one year using a single dose ring-vaccination strategy with delayed vaccination in the placebo-control group. Vaccine efficacy was high with protection established quickly [150]. In nonhuman primates 66% vaccine efficacy was seen just three days post vaccination [151]. Similar to disease severity, correlates of protection were defined as both IgG and IgM EBOV glycoprotein specific antibodies with NK cells conferring rapid protection and T follicular helper cells assisting the humoral response (reviewed in [152]).

### 4.5. Respiratory Syncytial Virus

RSV belongs to the *Pneumovirus* genus of the *Paramyxovirus* family, causing respiratory disease like the common cold with most children being infected by the age of 2. RSV can cause bronchiolitis and pneumonia in around one third of infants with a high risk (2.6%) of hospitalization at one month old [153]. Globally, there were over three million hospitalizations and 60,000 in-hospital deaths worldwide in 2015 [154]. Despite this significant disease burden, there is currently no licensed vaccine. Reasons for this include competition with maternal antibody in infants under six months [155], history of disease enhancement from a formalin-inactivated vaccine tested in the 1960s [156], and lack of protective immunity following infection [157]. Reinfection with RSV is not due to antigenic variation but rather viral immune evasion and a suboptimal immune response characterised by low levels of mucosal IgA memory [158], a high proportion of non-neutralizing antibody responses in primary infection [159], compromised T helper 1 response [160], and dendritic cell function [161], all of which correlate with disease severity during primary infection. Disease enhancement by the formalin-inactivated vaccine was characterized by an excess of lung eosinophils and pulmonary immune complex formation upon challenge in naïve children. Subsequent studies show the vaccine generated a T helper 2 bias due to the nature of the antigen and immune system immaturity. Moreover, an absence of MHC-I presentation of cytosolic antigen lead to reduced TLR stimulus and CD8+ T cell responses, a common pitfall in non-replicative vaccines [156]. These factors both contributed to reduced T cell help for elimination of pathogenic B cell clones during the process of affinity maturation (for a more detailed review on disease enhancement see [156]. The available data suggests a successful vaccine will need to elicit strong CD8+ T cell responses [162] coinciding with neutralising antibody [163] to ensure safety and adequate protection.

Higher viral loads at presentation are associated with less severe RSV disease [164,165,166], suggesting stimulation of the innate immune response early during infection predicts favourable disease outcomes. In addition, many single nucleotide polymorphisms linked to disease severity are in innate immunity genes [167]. Intriguingly a small study of another problematic respiratory virus, SARS-CoV-2, viral dynamics in mild and severe cases showed correlation between low viral load on presentation and severe disease [168]. Correlates of disease severity, such as robust initial IFN responses, should be used as a surrogate for ideal vaccine responses in clinical trials.

## 5. NSV Vaccines on the Market or under Development

Continued efforts are made to improve vaccine safety and efficacy involving use of innovative vaccine design platforms, such as use of new substrates for vaccine production, synthetic chemistry, replicating and non-replicating viral vectors, lipid nanoparticles, etc.

In this section, we will discuss NSV vaccines that are on the market (Table 1) or undergoing clinical trials (Table 2) with focus on the design strategies (Figure 2) that have been employed for their development.

### 5.1. Inactivated Virus Vaccines

Inactivated virus vaccines have been used for over a century, involving the use of ‘killed’ virus that is no longer infectious. For example, there are three types of inactivated influenza virus vaccines, the whole virus, split virus, and subunit. Once the whole virus is inactivated, it can be further disrupted with the use of surfactants to create a “split virus”. Further processing enables the separation of the HA and NA subunits from the other viral components creating the “subunit virus”. Additionally, using a protein expression system, a subunit vaccine can be made by producing recombinant vaccine protein in expression system cell lines. This is further discussed in the following section.

Their major advantage is safety, removing the possibility of virulence reversion compared to live vaccines. Whole virus and the split virus vaccines also offer the full range of viral antigens to the host-immune system unlike subunit or nucleic acid vaccines. Notably, studies have shown that split virus influenza vaccines had a greater clinical effectiveness than subunit ones [169], which might be due to a stronger cellular immune response induced in the split virus vaccines [170]. A major disadvantage, however, is the lack of virus replication that may limit the activation of the adaptive immune response, preventing generation of immunological memory. This type of platform has been widely used to produce NSV vaccines, such as IAV and RABV vaccines. RABV vaccines are produced either in primary culture of chicken fibroblasts (RabAvert), or in mammalian cells, such as the VERO cell line (VERORAB) or human diploid MRC-5 cell line (IMOVAX). Once the virus is propagated and harvested, it is inactivated by chemical agents such as β-propiolactone [171].

At present, the largest market of inactivated vaccines is for influenza. Different influenza virus strains are propagated separately in embryonated chicken eggs, inactivated with either a chemical agent (β-propiolactone or formaldehyde) or a combination of chemical and physical agents (ultraviolet light). These are then further chemically disrupted with sodium deoxycholate, or octylphenol ethoxylate to produce a “split virus” and finally combined together to produce trivalent or quadrivalent vaccine [172].

Another type of inactivated vaccines uses only the most immunogenic components of the virion and presents them to the immune system as a stable protein/s. These are termed subunit vaccines. The influenza subunit vaccines involve the surface antigens hemagglutinin (HA) and neuraminidase (NA). They are either purified from inactivated virus propagated in embryonated chicken eggs (AGRIFLU, FLUVIRIN), or MDCK cell lines (FLUCELVAX and AUDENZ, the first ever vaccine against the pandemic flu H5N1).

### 5.2. Live Virus

Live virus vaccines are one of the oldest vaccination strategies (the smallpox vaccine was introduced by Edward Jenner in 1796), using replication competent virus that has been attenuated through consecutive passages in tissue culture to no longer cause disease. The advantage of this strategy is that it mimics natural infection, enabling presentation of both surface and intracellular viral antigens and activation of both innate and the adaptive immune response to provide long-lasting immunity. However, it also raises additional safety concerns due to the potential for reversion to virulence. This type of strategy has been successful in measles, mumps, and influenza vaccination.

Measles and mumps vaccines were firstly introduced in the 1960s, and they have had a huge impact worldwide in disease morbidity and mortality. MV and (MuV) are included in the MMR (measles, mumps, varicella) vaccines such as M-M-R II^®^(Merck Sharp & Dohme, Chalfont, PA, USA), M-M-RVaxPro^®^ (Merck Sharp & Dohme), or in quadrivalent vaccines such as Proquad^®^ (Merck Sharp & Dohme) and Priorix-Tetra^®^ (GlaxoSmithKline Biologicals). The most used strains include Edmonston and Schwartz (measles) and Jeryl Lynn (mumps). These strains are propagated in chicken embryo cell cultures, then lyophilised and combined with either rubella virus (MMR II^®^, M-M-RVaxPro^®^) or rubella and varicella virus (Proquad^®^ and Priorix-Tetra^®^) to create the final vaccine.

There are two different types of live attenuated influenza vaccines (LAIV) commercially available, a quadrivalent vaccine, commercially known as FluMist (Table 1 and Figure 2), which came in the market in 2003 in USA, and a trivalent vaccine created by the Institute of Experimental Medicine, St Petersburg, Russia. The Russian vaccine has been in the market for more than 50 years [173,174] and is based on cold–adapted A/Leningrad/134/17/57 (H2N2) (Len–MDV) and B/USSR/60/69 Master Donor Viruses (MDVs) with the addition of a circulating seasonal strain [175]. The quadrivalent vaccine is based on the “Ann Arbour” backbone (A/Ann Arbor/6/60 and B/Ann Arbor/1/66) and contains two B strains (B/Yamagata/16/88 and the B/Victoria/2/87 lineages), in addition to a A/H1N1 and a A/H3N2 strain [176]. Each of the viruses is propagated in Specific pathogen-free (SPF) eggs, then blended together and diluted as required to attain the desired potency. This vaccine is administered via intranasal route, more closely resembling the natural infection, thus creating a broader immune response and avoiding the need of highly skilled personnel for its administration. The cold adapted strains used in LAIV vaccines can replicate only in the upper respiratory tract of the patients which makes them safer than other live vaccines. However, reduced vaccine efficacy has limited the use of LAIV vaccines [177].

### 5.3. Replication-Competent (Attenuated) Viral Vectors

An additional means to improve the safety of live vaccines is using a non-pathogenic virus as a vector. VSV belongs to the *Rhabdovirus* family and affects livestock, causing a rarely fatal disease with early symptoms similar to foot-and-mouth disease virus. Infection in humans is asymptomatic and there are extremely low levels of seroprevalence. Added to this, the genome can accommodate large inserts compared to alphavirus and poliovirus. Moreover, the VSV M protein inhibits the host immune response, thus favouring rapid production of viral proteins and high titre virus for vaccine manufacture [178]. This was used to develop the EBOV vaccine Ervebo, which uses the VSV virus to express the EBOV envelope glycoprotein in place of its own (Kikwit 1995 strain) [8]. This recombinant vaccine is grown in VERO cells (Table 1 and Figure 2) and is administered through the intramuscular route, eliciting a strong immune response [179] against the Ebola virus.

### 5.4. Replication-Deficient Viral Vectors

These defective viruses contain a mutation in an essential gene required for viral replication and assembly of the progeny. They are propagated in a cell line, which expresses the viral mutated gene, whereas in normal cells the viral genome is expressed but unable to produce functional virions [180]. Adenovirus (AdV) vectors are promising vaccine vectors for the delivery of viral antigens to the host target. Their ability to induce a transgene specific T cell response and accept large DNA transgenes and ease of growth at large scale have made this platform a promising candidate for vaccine delivery [181]. The deletion of the E1 gene from the virus and supplying it only in the packaging cell line (HEK293 or PER.C6) *in trans* renders this virus replication deficient. This further improves safety while maintaining the strong initial immune response of a live virus. Additional modification to this virus includes the deletion of the E1 gene, allowing for its replacement with an antigen gene of interests [181,182]. To minimise pre-existing immunity against the human Ad, new Ad vectors from non-human primates (NHP) have also been developed. One of the Ad vectors that is being tested in clinical trials includes the GSK’s Chimpanzee-derived Adenovector (ChAd155-RSV). This vector expresses the RSV F, N, and M2-1 proteins. This study will investigate on the safety and immunogenicity of the vaccine in infants (NCT03636906).

### 5.5. Virus-Like Particle (VLP) Vaccines

VLPs are particles composed commonly of virus envelope and/or capsid proteins but lack genetic material, making them non-infectious. VLP vaccines offer a multitude of advantages, such as safety, ease, and speed of synthesis, stability at room temperature, and ability to provide a broad immune response [183]. VLP vaccines have been used to immunize against filoviruses, whereby the glycoprotein (GP), matrix (VP40), and nucleocapsid (NP) peptides have been expressed in mammalian or insect cell lines, combined together in VLP vaccines and used for NHP immunization. Results from these studies have suggested that protection from EBOV strongly correlated with anti-GP IgG titers [184,185,186]. A recent phase I clinical trial showed promising results in testing an adjuvanted EBOV-GP VLP vaccine was able to elicit a robust and persistent immune response in humans [187].

This type of platform has also been used for production of vaccines against seasonal and highly pathogenic influenza H7N9 and H5N1 viruses. The highly pathogenic H7N9 VLP vaccine consisted of a combination of HA and NA of A/Anhui/1/13 with the matrix 1 protein (M1) of A/Indonesia/5/05. These proteins are produced in insect cells using a recombinant baculovirus [188].

Another seasonal influenza vaccine that uses the VLP platform is QVLP. Unlike other flu vaccines, it is produced in plants (*Nicotiana benthamiana*) using the Agroinfiltration transient expression platform [189] and consists of HA proteins from four different influenza strains, A/California/07/2009 H1N1 (A/H1N1 Cal), A/Victoria/361/11 H3N2 (A/H3N2 Vic), B/Brisbane/60/08 (B/Bris, Victoria lineage), or B/Wisconsin/1/10 (B/Wis, Yamagata lineage) [190,191]. This vaccine has recently completed phase 3 clinical trials with promising results.

### 5.6. Recombinant Subunits and Peptide Vaccines

Peptidic and recombinant subunits vaccines involve the use of specific proteins or peptides that are the primary antigenic targets for the virus, avoiding administration of live virus. These recombinant subunits are usually synthesized by using expression vectors. The use of recombinant and peptidic vaccines offers several advantages in term of safety, ease of production, lack of cold chain, and reduced costs. However, peptides are characterized by poor immunogenicity and they are susceptible to enzymatic degradation [192].

The only recombinant NSV vaccine on the market is Flublok. It is a quadrivalent vaccine composed of recombinant HA proteins from four influenza strains (A/Brisbane/02/2018 (H1N1), A/Kansas/14/2017 (H3N2), B/Maryland/15/2016, and B/Phuket/3073/2013). The HA proteins are synthesized in expresSF+^®^ insect cells using a baculovirus vector. The HAs are then purified by chromatography and blended together [107].

A recent study tested a synthetic peptide influenza vaccine (FLU-v) (Table 2) composed of four capsid and internal peptides, M1 protein, NP protein from A strain, NP protein from B strain, and M2 protein [119,193]. The proteins sequences were carefully selected for their T-cell reactive regions, and proteins were chemically synthesized [194]. FLU-v showed promising results in a phase IIb clinical trial, eliciting a cell mediated and humoral immune response to the vaccine antigens, and it recently completed a phase 3 clinical trial [107].

### 5.7. Nucleic Acid Vaccines

Nucleic acid vaccines involve the administration of DNA or RNA encoding virus antigen(s) to the recipient, utilising the cellular translation machinery to then produce viral proteins that elicit an immune response. Safety, stability of the product, ease of manufacturing, short time of production, and reduced costs are some of the advantages of this platform. Moreover, DNA vaccines have been shown to stimulate potent innate and adaptive immune responses in animal models [195]. However, this efficiency demonstrated in preclinical studies has not been translated to humans. To overcome this limitation, new delivery techniques involving electroporation and gene guns have been developed to improve DNA plasmid uptake [10,196,197].

A new generation of vaccines have been developed by using RNA lipid nanoparticles. A recent phase I clinical trial tested the safety and immunogenicity of a mRNA vaccine composed of a lipid nanoparticle, containing a chemically modified mRNA encoding for the hemagglutinin (HA) from H10N8 and H7N9 influenza [198]. Similar results were obtained for a combinational mRNA vaccine, mRNA-1653, encoding for the fusion protein of Human metapneumovirus (hMPV) and parainfluenza virus type 3 (PIV3) The vaccine induced a functional immune response in adults and is now being tested in infants (NCT04144348).

## 6. Challenges to Vaccine Manufacturing

Vaccine manufacturing is an incredibly complex and challenging industry, due in part to the costs associated with product development, capital expenditure, direct cost of goods (COGs), as well as licensure and marketing. The industry is continually looking at mechanisms to increase output and/or decrease cost, which would not only allow for increased profits but also increased global availability and improved health outcomes. As many NSV vaccines are manufactured in cells/eggs from higher order animals, the host immune response may play a critical role in manufacturing limitations by decreasing viral growth and ultimately increasing the COGs.

Most viral vaccines are grown in either primary cell cultures, diploid, or continuous cell lines, or in some cases, embryonated eggs are used as the substrate. Influenza vaccines have been grown in eggs since the 1940s, and as of 2017, >400 million doses of trivalent vaccine were produced each year [199]. Given each dose takes between 1–2 eggs to produce, this would equate to 400–800 million eggs. This large number of highly specialized eggs is a serious burden to the manufacturing industry and any developments aimed at improving production systems would be of significant advantage. A considerable body of work has gone into the development of cell-based influenza production, particularly in MDCK cells, as well as recombinant influenza production, with the intention of improving manufacturing flexibility and vaccine efficacy [200]. In addition to the use of ‘clean eggs’, that is eggs that are held at higher biosecurity and improved cleaning conditions, for influenza vaccine growth, measles and mumps vaccine production utilizes cells derived from specific pathogen free (SPF) eggs as their substrate, although as with many vaccines, the use of continuous cell lines continues to be explored [201]. Rabies vaccines are highly effective; however, their production can be cost prohibitive for communities with the highest rates of rabies infections and deaths [202]. Rabies vaccines were originally derived from infected nerve tissue; however, with the advent of tissue culture, primary hamster cells [203], human diploid Wi-38 and MRC-5 cells [204], VERO cells [205] and chicken embryo cells [206] are now used. With the recent advent and licensing of the VSV-EBOV vaccine [8], additional work continues to improve yield in VERO cells as cost and manufacturing scale are critical in areas most impacted by Ebola [207]. Given that these manufacturing systems use immunocompetent cells for their derivation, understanding how the host immune system may impact production, cost, and even the vaccine itself may lead to improvements in production processes.

It has been demonstrated that embryonated eggs can mount an innate immune response to influenza, mediated through IL-1β and type I IFNs [208]. Bassano et al., (2013) demonstrated that IFN induced transmembrane proteins (IFITMs) also play a role in inhibition of influenza growth in chicken cells, suggesting an IFITM deleted egg may be enhanced for influenza vaccine production [209]. It is reasonable to assume that inhibition of such innate molecules could ultimately increase vaccine production. It was demonstrated some time ago that VERO cells were unable to produce type I IFNs and were not sensitive to the synthetic viral nucleic acid analogue poly-(I:C) [210]. This may explain their high level of use in viral vaccine production, as a key innate regulator is not present, allowing high levels of infection. Further investigation into the host immune response to viral vaccines has been undertaken using genome-wide screens in VERO cells to identify host factors limiting vaccine growth [211,212]. In these studies, the positive sense RNA viruses poliovirus and rotavirus were used to identify multiple genes relating to cell cycle progression, proliferation, and apoptosis that inhibited viral replication. Wang et al. (2019) demonstrated that rabies vaccine growth in MRC-5 cells resulted in upregulation of miR-423-5p, which ultimately increased IFNβ expression through reduction of suppressor of cytokine signalling 3 (SOCS3) [144], the consequence of which was reduced viral production. Therefore, having a better understanding of the host pathogen interaction between NSV and the vaccine substrates may provide a mechanism for developing improved vaccine manufacturing processes. Clearly, future efforts should focus on delivering in this area.

## 7. Conclusions

The exponential scientific improvements we are observing in the field of vaccination are rapidly bringing us to a world in which a healthy existence is a realistic possibility for everyone. We are now well positioned to study the complex associations between infectious agents and their hosts, informing novel vaccine strategies and detection systems to enable effective control of viral spread. Nevertheless, the recent influenza and coronavirus pandemics have illustrated the need for both cost-effective and rapidly developed vaccine solutions on a global scale. Novel, technology-enabled substitutes to our current methods for developing innovative vaccine solutions are essential to solve some of the most difficult global health challenges with regards to NSV. Researchers focused on the host–pathogen interface are uniquely placed to contribute novel and radical innovations in dealing with NSV, particularly in the areas of vaccination. The challenge is to continue the highly valued efforts in the management of NSV infectious diseases by improving vaccination and enhancing conventional systems and additionally build new approaches.

Vaccines against NSVs pose difficulties due to their myriad host immune evasion abilities, and the peculiarities of each host–virus interaction and subsequent diseases. As such, each vaccine approach has to be tailored specifically on an individual basis. The more we know about NSVs host–pathogen exchanges and immune modifying mechanisms, the better and more efficacious vaccines will become. Therefore, it is essential to utilize the ever-expanding viral technologies available such as Adenoviruses, VLPs, nucleic acid vaccines, or improved cell lines substrates to enhance current vaccine development methods for NSVs. Clearly, there are excellent opportunities to leverage the immunology associated with the host–pathogen interface in combination with recent technological improvements to improve vaccine development and vaccine manufacture for NSVs. By actively pursuing research into the host–pathogen interaction, we can link up and synergistically enhance NSV vaccination approaches.

## Figures and Tables

**Figure 1 vaccines-09-00059-f001:**
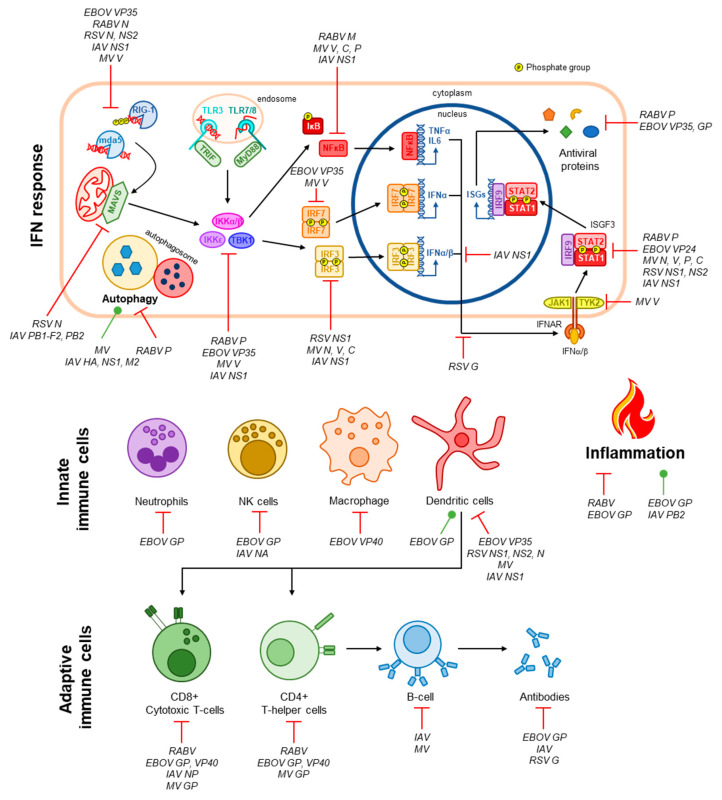
Immune evasion by NSVs. The IFN response is the primary antiviral pathway activated following virus infection. NSVs are detected by intracellular PRRs such as RIG-1 and mda5 in the cytoplasm or TLR3/7/8 in the endosomal compartment via interactions with viral RNA [12]. Signalling cascades involving MAVS (for RIG-I and mda5) or TRIF/MyD88 (for TLRs) result in the activation of the kinases IKKα/β/ε and TBK1 that subsequently activate the latent transcription factors IRF3, IRF7, and NFκB via phosphorylation [12]. These transcription factors promote the expression and secretion of proinflammatory and antiviral cytokines (TNFα, IL-6, and IFNα/β) [12]. IFNs signal in an autocrine and paracrine manner by binding to the IFNα/β receptor (IFNAR), resulting in the activation of the ISGF3 complex (STAT1, STAT2, and IRF9) through phosphorylation of STAT1 and STAT2 by IFNAR-associated kinases JAK1 and TYK2 [12]. ISGF3 then promotes the expression of antiviral genes [12]. Innate and adaptive immune cells, as well as inflammation and autophagy, which additionally contribute to viral clearance, are also depicted. Targeting of the antiviral immune response by NSVs, and the viral proteins involved if known, are indicated [33,34,35,36,37,38,39,40,41,42,43,44,45,46,47,48,49,50,51].

**Figure 2 vaccines-09-00059-f002:**
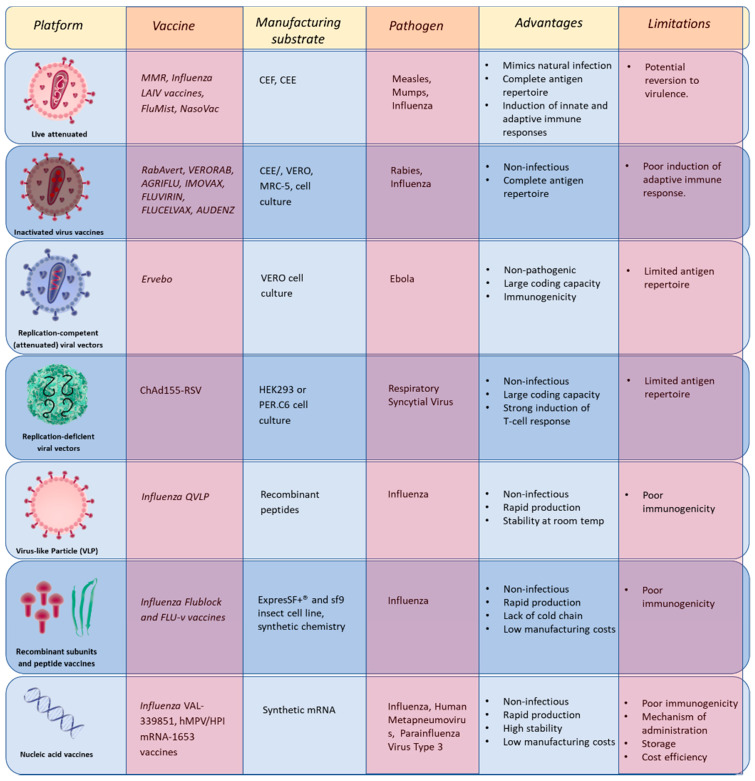
Vaccine design platforms used for the production of NSV vaccines. Description of platforms used to develop NSV vaccines, including examples, advantages, and limitations. CEF = Chicken Embryo Fibroblasts; CEE = Chicken Embryonated Eggs; expresSF+ and sf9 = *Spodoptera frugiperda* insect cell lines using a baculovirus expression vector system; HEK293 = Human Embryonic Kidney cells; PER.C6 = Human Embryonic Retinoblasts; VERO = African green monkey kidney cell line; MRC-5 = human fetal lung strain MRC-5; RSV = Respiratory Syncytial Virus; (ChAd155-RSV) = Chimpanzee-derived Adenovector harboring specific RSV proteins; hMPV/PIV = human metapneumovirus and parainfluenza virus type 3.

**Table 1 vaccines-09-00059-t001:** List of NSV vaccines in the market.

Vaccine Name	Agent	Manufacturer	Approved Date	Manufacturing Platform	Live/Inactivated	Route of Administration
Ervebo	Ebola Zaire Vaccine	Merck Sharp & Dohme	2019	Cell culture (VERO)	VSV attenuated virus	Intramuscular injection
AFLURIA^®^ QUADRIVALENT	Influenza A and B	Seqirus	2016	Chicken Embryonated Eggs	Inactivated	Intramuscular injection
AFLURIA^®^	Influenza A and B	Seqirus	2007	Chicken Embryonated Eggs	Inactivated	Intramuscular injection
Fluzone	Influenza A and B	Sanofi Pasteur	1980	Chicken Embryonated Eggs	Inactivated	Intramuscular injection
Fluzone High-Dose	Influenza A and B	Sanofi Pasteur	2009	Chicken Embryonated Eggs	Inactivated	Intramuscular injection
Fluzone Intradermal Quadrivalent	Influenza A and B	Sanofi Pasteur	2014	Chicken Embryonated Eggs	Inactivated	Intramuscular injection
Fluzone Quadrivalent Southern Hemisphere	Influenza A and B	Sanofi Pasteur	2013	Chicken Embryonated Eggs	Inactivated	Intramuscular injection
FLUARIX	Influenza A and B	GlaxoSmithKline	2005	Chicken Embryonated Eggs	Inactivated	Intramuscular injection
FLUARIX QUADRIVALENT	Influenza A and B	GlaxoSmithKline	2012	Chicken Embryonated Eggs	Inactivated	Intramuscular injection
FLULAVAL	Influenza A and B	ID Biomedical Corporation of Quebec	2006	Chicken Embryonated Eggs	Inactivated	Intramuscular injection
FLULAVAL QUADRIVALENT	Influenza A and B	ID Biomedical Corporation of Quebec	2013	Chicken Embryonated Eggs	Inactivated	Intramuscular injection
Ultrix Quadri	Influenza A and B	Nacimbio	2019	Chicken Embryonated Eggs	Inactivated	Intramuscular injection
FLUCELVAX QUADRIVALENT	Influenza A and B	Seqirus	2016	Cell culture (MDCK)	Inactivated-Subunit	Intramuscular injection
Flucelvax	Influenza A and B	Seqirus	2012	Cell culture (MDCK)	Inactivated-Subunit	Intramuscular injection
AUDENZ	Influenza A (H5N1)	Seqirus	2020	Cell culture (MDCK)	Inactivated-Subunit	Intramuscular injection
AGRIFLU	Influenza A and B	Seqirus	2009	Chicken Embryonated Eggs	Inactivated-Subunit	Intramuscular injection
FLUVIRIN^®^	Influenza A and B	Seqirus	1988	Chicken Embryonated Eggs	Inactivated-Subunit	Intramuscular injection
Flublok	Influenza A and B	Protein Sciences Corporation/Sanofi Pasteur	2013	Cell culture (expres sf9)	Recombinant inactivated	Intramuscular injection
Flublok Quadrivalent	Influenza A and B	Protein Sciences Corporation/Sanofi Pasteur	2013	Cell culture (expres sf9)	Recombinant inactivated	Intramuscular injection
FluMist	Influenza A and B	MedImmune/AstraZeneca	2003	Chicken Embryonated Eggs	Live attenuated virus	Intranasal
NasoVac-S	Influenza A and B	Serum Institute of India	2019	Chicken Embryonated Eggs	Live attenuated virus	Intranasal
M-M-R II	Measles, Mumps, and Rubella Virus Vaccine	Merck Sharp & Dohme	2008	MV and MuV from Chicken Embryo cell culture/RV from WI-38 human diploid lung cells	Live attenuated virus	Subcutaneous injection
Proquad	Measles, Mumps, Rubella and Varicella Virus Vaccine	Merck Sharp & Dohme	2005	MV and MuV from Chicken Embryo cell culture/RV and VV from respectively WI-38 and MRC-5 human diploid lung cells.	Live attenuated virus	Subcutaneous injection
M-M-RVAXPRO	Measles, Mumps and Rubella Virus Vaccine	Merck Sharp & Dohme	2006	MV and MuV from Chicken Embryo cell culture, RV from WI-38 human diploid lung cells	Live attenuated virus	Intramuscular or subcutaneous injection
PRIORIX-TETRA	Measles, Mumps, Rubella and Varicella Virus Vaccine	GlaxoSmithKline Biologicals	2005	MV and MuV from Chicken Embryo cell culture; RV and VV from MRC-5 human diploid lung cells	Live attenuated virus	Subcutaneous injection
Imovax	Rabies	Sanofi Pasteur	2011	Cell culture (MRC-5)	Inactivated	Intramuscular injection
VERORAB	Rabies	Sanofi Pasteur	2005	Cell culture (VERO)	Inactivated	Intramuscular injection
RabAvert	Rabies	GlaxoSmithKline	1997	Chicken Embryo cell culture	Inactivated	Intramuscular injection

VERO = African green monkey kidney cell line; MDCK = Madin-Darby Canine Kidney cells; expresSF+ and sf9 = *Spodoptera frugiperda* insect cell lines using a baculovirus expression vector system; MV = measles virus; MuV = mumps virus; RV = rubella virus; VV = varicella virus; WI-38 = Winstar Institute 38 cell line from human diploid lung fibroblasts; MRC-5 = human fetal lung strain cells.

**Table 2 vaccines-09-00059-t002:** List of NSV vaccines undergoing clinical trials.

Vaccines in Clinical Trials	Agent	Manufacturer	Approved Date	Manufacturing Platform	Live/Inactivated	Route of AdminIstration	Trial Number
Purified Vero Rabies Vaccine—Serum Free Vaccine generation 2 (VRVg-2)	Rabies	Sanofi Pasteur	2019	Cell culture (VERO)	Inactivated	Intramuscular injection	NCT03965962
ChAd155-RSV vaccine	RSV	GlaxoSmithKline	2019	Cell culture (HEK 293, PER.C6)	Live	Intramuscular injection	NCT03636906
mRNA-1653	hMPV/HPI	Moderna	2019	mRNA-Lipid Nanoparticle	Synthetic mRNA	Intramuscular injection	NCT04144348
VAL-339851	Influenza A (H7N9)	Moderna	2016	mRNA-Lipid Nanoparticle	Synthetic mRNA	Intramuscular injection	NCT03345043
QVLP	Influenza A and B	Medicago	2017	Recombinant peptides	Virus-Like Particles	Intramuscular injection	NCT03301051
FLU-v	Influenza A and B	PepTcell (SEEK)	2017	Synthetic peptides	Fmoc synthetic chemistry	Subcutaneous injection	NCT03301051
EBOV GP Vaccine	Ebola	Novavax	2015	Baculovirus/Sf9 cells	Recombinant GP protein	Intramuscular injection	NCT02370589

HEK 293 = Human Embryonic Kidney cells; PER.C6 = Human Embryonic Retinoblasts; hMPV/PIV = human metapneumovirus and parainfluenza virus type 3; sf9 = *Spodoptera frugiperda* insect cell lines using a baculovirus expression vector system.

## Data Availability

The data presented in this study are collected from the cited literature.

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
