# Peer review of "Investigating the Interaction between Negative Strand RNA Viruses and Their Hosts for Enhanced Vaccine Development and Production"

_vaccines, 2021, doi:10.3390/vaccines9010059_

Round 1

Reviewer 1 Report

The Authors have fully addressed my previous comments. The manuscript is suitable for pubblication

Reviewer 2 Report

Manuscript by Mara et al. entitled as, "Investigating the interaction between negative strand RNA viruses and
their hosts for enhanced vaccine development and production" is a review article. The manuscript has comprehnsively covered various important aspects related to vaccine design against negative strand viruses such as Ebola, Influenza, Measels etc. of greater medical relevance. Authors have very nicely presented host-pathogen interactions and intricacies of immune system invovled. 

Overall, manuscript is very well prepared and is acceptable for publication.

It should be checked for typos, e.g. line 353 has "c.$.." against dollar values. 

This manuscript is a resubmission of an earlier submission. The following is a list of the peer review reports and author responses from that submission.

Round 1

Reviewer 1 Report

This manuscript by Mara et al. is an updated review of the literature regarding vaccines for the main negative strand RNA viruses causing diseases in humans. In the era of SARS-CoV-2 pandemics this is indeed a very interesting topic. However, in the opinion of this Reviewer some issues need to be addressed before publication, as reported below:

  • Taking into account the title of this review, one would expect to find a different organization of the text. Indeed, current chapter 4 and 5 should be the most relevant ones. I would like to mention that looking at the impact of viral/host interplay on the success or failure of a vaccine is an original and very interesting way to present the topic and thus I would strongly recommend the Authors to work on the organization of the review giving the right light this aspect. Maybe chapters 4 and 5 could be further enriched while chapters 1 to 3 could be reduced and summarized.
  • The introduction is too repetitive and some of the sentences are written in a too "conversational" tone. I would suggest a deep revision of the language throughout the text
  • Figure 1 (very interesting) and Table 1 should belong to a full chapter with a description of what it is displayed
  • I would better clarify what is a "inactivated virus vaccine" with respect to a "subunit vaccine", a "split vaccine" and so on, as currently there is a bit of confusion.
  • In the opinion of this Referee it would greatly increase the soundness of this manuscript a mention to vaccination strategies developed or under development for Coronaviruses in general and SARS, MERSV and SARS-CoV-2 in particular.

Reviewer 2 Report

This is an excellent overview summarizing current achievements in design and development of NSV vaccines. It is original, comprehensive and well illustrated.

Compliments to the authors.

Two minor comments please have a look at Line 498 -499, it is unclear why this is stated here. In the conclusions Line 637-638 I like to positive forward outlook and I am sure this statement will be supported by many rich countries but I doubt if it is truly the case taken into the costs, supply routes and general living conditions in many other parts of the globe.

Reviewer 3 Report

The manuscript submitted in the form of a review article by Kostlend et al, entitled as, “Investigating the interaction between negative strand viruses and their hosts for enhanced vaccine development and production” compiles information on vaccines against several negative strand RNA viruses. Authors have attempted to write this manuscript in the backdrop of COVID-19 pandemic, however it is poorly prepared and does not aid in our present understanding of vaccine related challenges in general, and COVID-19 in particular. Authors need to go through structural organization of the manuscript, flow of the story, rationale of the topic, and include relevant content to make the article appealing to the target audience.

Major Comments:

  1. The title of the manuscript should mention “negative strand RNA viruses”.
  2. Keywords should also include RNA virus. Keyword “NSV vaccines” generates only 930 results in google scholar. Authors should put more thoughts in selecting keywords.
  3. The first line of abstract describes COVID-19, driving curiosity of the readers; however, in the entire manuscript COVID-19 is mentioned only two times, and SARS only once. Thus, disappointing readers.
  4. The introduction section does not setup reader that what they should be expecting in the manuscript. Describe how NSVs are different from positive strand RNA viruses, DNA viruses, and why studying them is even more challenging e.g. lack of animal models or cell lines etc. What is the rationale for writing this manuscript is not clear in the introduction section. Words like “significant impact on human health” are not definitive, rather authors should support their statements with numbers and citation.
  5. It is not clear to this reviewer, what criteria was used in the selection of detailed viruses in the manuscript. What is the import of discussing these viruses here. The VSV mainly impacts animals and mostly causes asymptomatic infection in human, except some livestock owners.
  6. Manuscript need consideration on the organization of the content. The introduction should describe what is NSV, why they are important, what we know about them, what are the challenges, and what are the remaining gaps. Next should be details of individual viruses including their disease pathogenesis, host-pathogen interaction, life cycle, methods to control the infection, successes achieved, knowledge gaps, and how those can be fulfilled. Third section should be various kinds of vaccines that can designed, successes achieved, knowledge gaps, and how those can be fulfilled. Finally, in the backdrop of this information, details of vaccines available, why they need improvement etc. should have been described. This rationale should have been justified in the vaccine which are under development/clinical trial. The conclusion section needs to provide a good take home message for the readers.
    While most of points mentioned above are included, but they are not making a coherent story. It is lacking a flow and reader need to paddle through the sea of the information.
  7. Authors should pay special attention to proof-read the manuscript. Lines 498-500 seems to be a comment and inadvertently included in the main text.

Minor comments:

In Table 1, vaccines M-M-RVAXPRO and PRIORIX-TETRA, lacks several key details.

Vaccines in clinical trial could be a separate table.

Figure 1 is not helpful in explaining intricacies of the various types of vaccines. It may have points detailing salient features of every type, advantages, disadvantages, and examples.